# The Spatial and Temporal Distribution of Process Gases within the Biowaste Compost

**Sylwia Stegenta** [1,*] , **Karolina Sobieraj** [1], **Grzegorz Pilarski** [2], **Jacek A. Koziel** [3]
**and Andrzej Białowiec** [1,*]

[1]   Faculty of Life Sciences and Technology, Wroclaw University of Environmental and Life Sciences, 37a
      Chełmońskiego Str., 51-630 Wrocław, Poland; karolina.sobieraj@upwr.edu.pl
[2]   Best-Eko Sp. z o.o., 1 Gwarków Str., 44-240 Żory, Poland; grzegorz.pilarski@best-eko.pl
[3]   Iowa State University, Department of Agricultural and Biosystems Engineering, IA 50011-3270, USA;
      koziel@iastate.edu
*     Correspondence: sylwia.stegenta@upwr.edu.pl (S.S.); andrzej.bialowiec@upwr.edu.pl (A.B.)

**Abstract:** Composting is generally accepted as the sustainable recycling of biowaste into a useful and beneficial product for soil. However, composting processes can produce gases that are considered air pollutants. In this dataset, we summarized the spatial and temporal distribution of process gases (including rarely reported carbon monoxide, CO) generated inside full-scale composting piles. In total 1375 cross-sections were made and presented in 230 figures. The research aimed to investigate the phenomenon of gas evolution during the composting of biowaste depending on the pile turning regime (no turning, turning once a week, and turning twice a week) and pile location (outdoors, and indoors in a composting hall). The analyzed biowaste (a mixture of tree leaves and branches, grass clippings, and sewage sludge) were composted in six piles with passive aeration including additional turning at a municipal composting plant. The chemical composition and temperature of process gases within each pile were analyzed weekly for ~49–56 days. The variations in the degree of pile aeration ($O_2$ content), temperature, and the spatial distribution of CO, $CO_2$ and NO concentration during the subsequent measurement cycles were summarized and visualized. The lowest $O_2$ concentrations were associated with the central (core) part of the pile. Similarly, an increase in CO content in the pile core sections was found, which may indicate that CO is oxidized in the upper layer of composting piles. Higher CO and $CO_2$ concentrations and temperature were also observed in the summer season, especially on the south side of piles located outdoors. The most varied results were for the NO concentrations that occurred in all conditions. The dataset was used by the composting plant operator for more sustainable management. Specifically, the dataset allowed us to make recommendations to minimize the environmental impact of composting operations and to lower the risk of worker exposure to CO. The new procedure is as follows: turning of biowaste twice a week for the first two weeks, followed by turning once a week for the next two weeks. Turning is not necessary after four weeks of the process. The recommended surface-to-volume ratio of a compost pile should not exceed 2.5. Compost piles should be constructed with a surface-to-volume ratio of less than 2 in autumn and early spring when low ambient temperatures are common.

**Dataset:** Submitted as the supplementary file at: https://www.mdpi.com/2306-5729/4/1/37/s1.

**Dataset License:** CC-BY

**Keywords:** biomass; agricultural residues; organic waste; sewage sludge; municipal waste; waste management; composting; aeration; emissions; carbon monoxide; greenhouse gases; carbon cycle; nitrogen cycle

## 1. Summary

Limited public awareness and the recent increase in consumption are contributing to the excessive exploitation of natural resources and modification of the environment, which includes the production of larger amounts of biowaste. The term 'biowaste' encompasses a wide variety of bio-based materials such as the organic fraction of municipal solid waste, yard waste (grass clippings, leaves, and wood residues), and sewage sludge. All these are abundant, especially in urban settings, and can present challenges for proper collection, treatment, and disposal. The most common method of biowaste treatment is composting. Composting processes result in the reduction of volume and weight of waste intended for final disposal (e.g., landfill). Composting can also be used to produce organic fertilizer. Current EU goals aim to phase out landfill and instead focus on 'zero waste' and 'circular economy' objectives consistent with sustainable development. Thus, policy pressure will increasingly be applied to improve and optimize common biowaste composting practices to align with the above goals.

Gaseous emissions from compost operations are important for several reasons. They can impact local and regional air quality via emissions of volatile organic compounds (VOCs) and odor. Stegenta et al. [1] reported that full-scale municipal waste composting operations release other air pollutants such as $CO$, $CO_2$, $NH_3$, $NO$, $NO_2$, $SO_2$, and $H_2S$, counted as greenhouse gases, odors, and fine particulate matter (PM2.5) precursors. Thus, it is reasonable to consider spatial and temporal variations of process gases that could be mitigated to lower the risk of worker exposure and for more sustainable composting. The chimney effect and its influence on thermal conditions and the degree of aeration have been studied [2,3]. The presence of $CO$ in gases emitted from composting piles has been reported [4–8]. The distribution of $CO$ in composting piles and the influence of the turning regime on $CO$ formation has not yet been studied. Gases generated inside compost can also provide useful information about the status of the biomass decay process and could potentially be exploited to optimize and control the process. The concept of using compost gases for the non-invasive, biosecure monitoring of decay process completion has been used for disposal of potentially infectious animal carcasses [9–15].

The quantitative detection of gases emitted from fugitive and area sources (such as composting operations) to the environment is important to develop, optimize and implement composting technologies, and improve emission models and air quality inventories. To date, relatively little is known about gaseous emissions (especially $CO$) from composting, specifically in systems with passive aeration. We hypothesized that the $CO$ spatial distribution depends on the composting pile turning regime and is correlated with temperature, $O_2$, and $CO_2$ distribution. We also hypothesized that the measurement of the temporal and spatial variations of temperature and process gases might be a useful tool for observing the organic matter stabilization. In addition, the resulting dataset may enable the optimization of biowaste composting, which increases the rate of organic matter decomposition while lowering process gas generation.

Therefore, the research aimed to investigate the phenomenon of gas evolution during the composting of biowaste depending on the pile turning regime (no turning, turning once a week, and turning twice a week), and pile location (outdoors and inside a composting hall). The analyzed biowaste (a mixture of tree leaves and branches, grass clippings, and sewage sludge) were composted in six piles with passive aeration. The chemical composition and temperature of the process gases of each pile were analyzed weekly for ~49–56 days. The variations in the degree of pile aeration ($O_2$ content), temperature, and the spatial distribution of $CO$, $CO_2$ and $NO$ concentrations during the subsequent measurement cycles were summarized and visualized.

The emission of $NO$ was the most varied and occurred in all conditions. There was no noticeable impact of turning and no apparent trend towards greater gas generation in larger piles. The lowest $O_2$ concentrations were associated with the central (core) part of the pile. Higher $CO$ and $CO_2$ concentrations and temperature were observed in the summer season, especially on the south side of piles located outdoors.

The dataset (consisting of 230 figures, and the raw data in the Excel spreadsheet in the supplementary materials) highlights the effect of different turning regimes and pile location on

the spatial and temporal distribution of $O_2$, CO, $CO_2$, NO and temperature within the compost. The dataset can also allow a more accurate understanding of the N and C transformations in the composted material, which in turn can be potentially explored to mitigate gaseous emissions from the process. The obtained data can be used to derive some technological recommendations for composting plant management. Examples of recommendations include:

(a) The frequency of turning for a particular phase of composting.
(b) The volume-to-surface ratio for different environmental conditions (e.g., ambient temperature, seasons)

This research can help in developing improved and environmentally-friendly methods for composting process management. The data may be relevant for composting optimization and further research that focuses on increasing the efficiency of the composting process.

Two main limitations associated with data acquisition and processing were identified. First, the measurements were time- and labor-consuming due to the full-scale size of the composting piles and the logistics of working at a large composting plant. Only two piles were examined per day by one three-member team. One possible solution to overcome this problem is to budget for additional properly-trained teams with analyzers that could be collecting data simultaneously. Another possible solution is to perform statistical analyses to determine if it is possible to limit the density and the number of samples collected and still represent the spatial and temporal variations of the composting process. The permanent installation of stainless steel probes is not possible due to the turning of the piles. The second limitation was the labor-consuming data processing, namely transferring the data to the Surfer software. More user-friendly software allowing dynamic modeling and visualization of the spatial distribution of temperature and gases in the composting piles within the timeframe in both 2D and 3D modes would be useful. It would bring new insights into the process monitoring and optimization.

Further full-scale research on the temperature and the spatial distribution of gases, especially with regard to the influence of the operational regime—turning frequency, pile size, pile moisture content, and waste type (biodegradability)—on CO (as a primary air pollutant) should be continued. The next experiment should include the measurement of additional gases—$CH_4$, $NH_3$, $H_2S$, $N_2O$, and VOCs—to produce a better understanding of the C and N cycle. In addition, the microbial characteristics and activity (e.g., the biodiversity of microbial population dynamics, respiration activity, and carbon monoxide dehydrogenase expression) should be studied. Finally, the compost matrix measurements should be complemented by surface temperature and gas emission distribution measurements. This would allow the creation of a holistic model for C and N in compost. It could allow the building of a multicriteria model for the optimization of biowaste composting to increase the organic matter decay rate while mitigating pollutant emissions.

## 2. Data Description

An example of the data visualization is provided in Figure 1. The figures presented in the supplementary materials (Figures.zip) summarize visualizations of the degree of pile aeration ($O_2$ concentrations), temperature, CO, $CO_2$, and NO concentrations and their spatial distribution variations during measurement cycles. For each of the six piles, four cross-sections and two longitudinal cross-sections (left and right side of the pile) were made. In total 1375 cross-sections were made and are presented in 230 figures. Figures S1–S46 summarize the spatial and temporal distribution of temperature in piles S1–S6. Figures S47–S92 summarize the spatial and temporal distribution of $O_2$. Figures S93–S138 summarize the spatial and temporal distribution of $CO_2$ concentrations. Figures S139–S184 summarize the spatial and temporal distribution of CO. Figures S185–S230 summarize the spatial and temporal distribution of NO. The raw data are also summarized in a separate file (Table S1.xlsx).

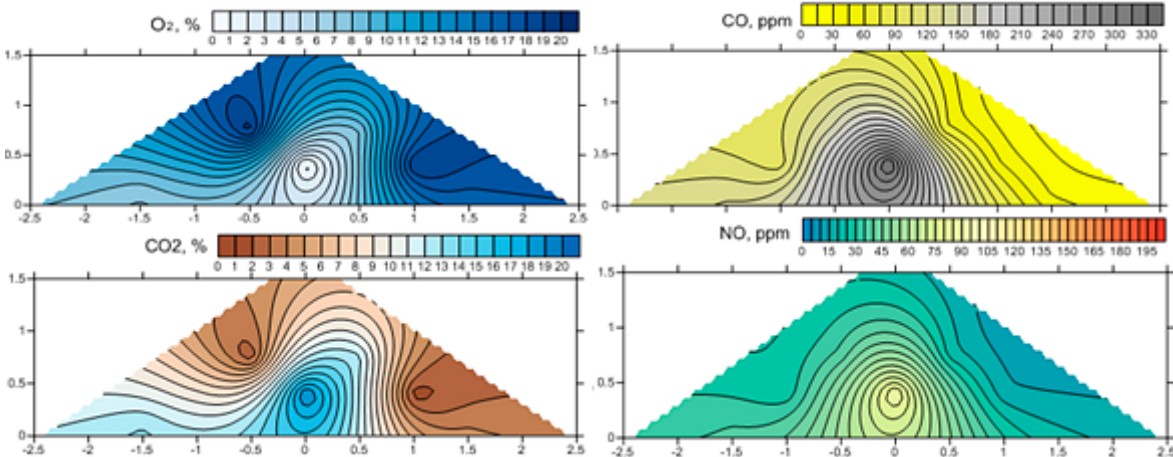

**Figure 1.** Example of the spatial distribution of $O_2$, CO, $CO_2$, NO. Cross-sectional areas of the monitored compost piles are presented in 230 figures summarizing the spatial and temporal distributions of gas concentrations and temperatures.

## 3. Methods

The evolution of gas and temperature (analyzed using a Kigaz 300 provided by Kimo Instruments, Chevry-Cossigny, France) during the composting of biowaste depending on the pile turning regime (no turning, turning once a week, and turning twice a week) (Figure 2), and pile location (outdoors and inside a composting hall) (Figure 3) was investigated.

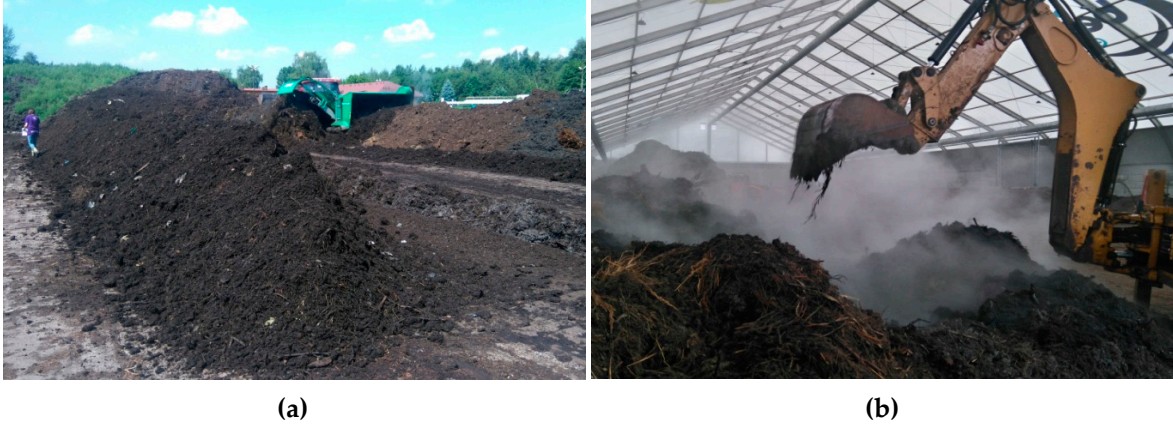

| (a) | (b) |

**Figure 2.** Turning of the compost material (**a**) pile A2 outdoors; (**b**) pile A5 inside a composting hall.

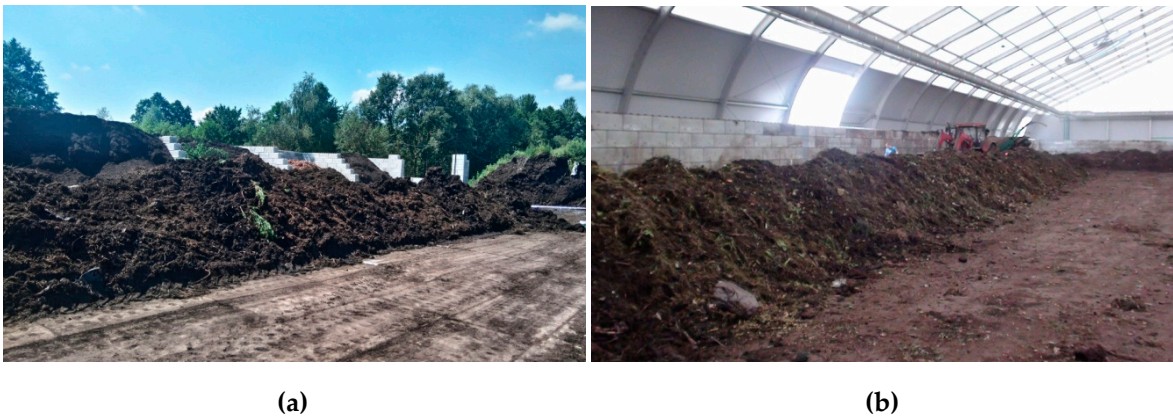

| (a) | (b) |

**Figure 3.** Composted biowaste at the beginning of the process in piles located (**a**) outdoors; (**b**) indoors in the composting hall.

The analyzed biowaste (a mixture of tree leaves and branches, grass clippings, and sewage sludge in volumetric proportion 4:2:1, respectively) were composted in six piles with passive aeration including additional turning. The experimental design and methods used to acquire the data, such as the characteristics of the composted waste, physicochemical properties of the pile components, the dimensions of the tested piles, placement of measurement points in the piles and detailed information of the test configurations are presented in the research article [16]. Briefly, the experimental design is presented in Table 1.

**Table 1.** The configuration of the experiment design, data acquisition cycles and a number of collected gaseous concentration samples and temperature measurements (Measurements were conducted from July to December, 2017).

| Pile | Process Time, Days | Turning Regime | Pile Location | Number of Temperature and Gaseous Sampling Cycles | Number of Sampling Cross-Sections | Number of Sampling Points in Each Cross-Section | Number of Collected Samples of Temperature and Gas |
|---|---|---|---|---|---|---|---|
| A1 | 50 | 2 times a week | Outdoor | 8 | 4 | 7 | 224 |
| A2 | 57 | 1 time a week | Outdoor | 8 | 4 | 7 | 224 |
| A3 | 52 | None | Indoor | 8 | 4 | 7 | 224 |
| A4 | 52 | None | Outdoor | 8 | 4 | 7 | 224 |
| A5 | 50 | 1 time a week | Outdoor/indoor | 8 | 4 | 7 | 224 |
| A6 | 50 | 1 time a week | Indoor | 8 | 4 | 7 | 224 |
| Total | 311 | - | - | 48 | 4 | 7 | 1344 |

The chemical composition and temperature of the process gases in each pile were analyzed weekly for ~49–56 days with the application of a stainless steel probe capable of collecting and analyzing gas samples from within piles. A flue gas analyzer Kigaz 300 by (Kimo Instruments, Chevry-Cossigny, France) equipped with electrochemical sensors ($O_2$, $CO_2$, CO, and NO) and thermocouple was used (Figure 4).

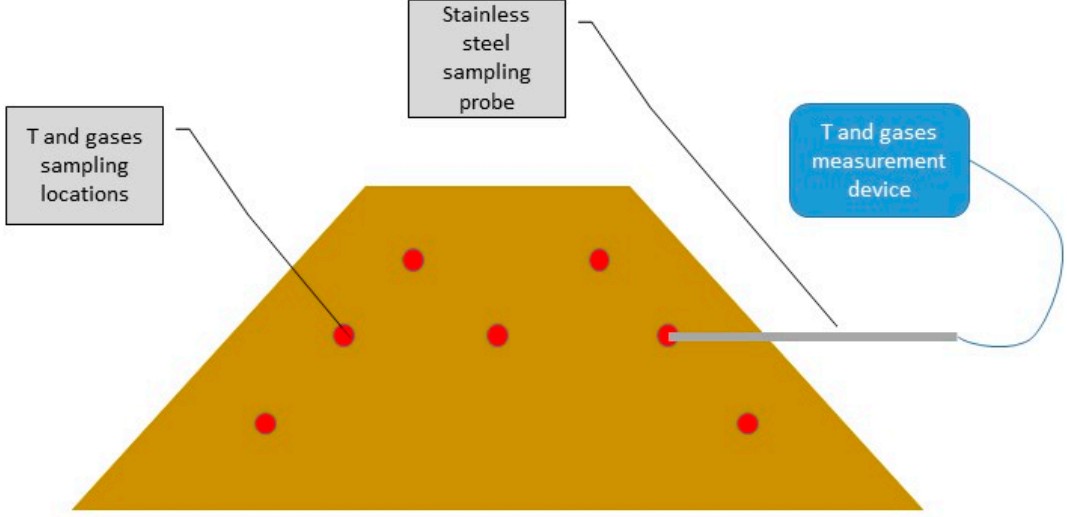

**Figure 4.** Gas ($O_2$, CO, $CO_2$, NO) concentrations and temperature measurements at a full-size composting operation processing biowaste (a mixture of leaves, branches, grass clippings, and sewage sludge). A perforated stainless steel gas probe was introduced into a pile at predetermined depths and cross-sections. Red points show the position of gas and temperature sampling points. The schematic probe for measuring the chemical composition of the gas and temperatures in a pile combined with the analyzer unit is provided.

The variations in the degree of pile aeration ($O_2$ content), temperature, and the spatial distribution of CO, $CO_2$ and NO concentration during the subsequent measurement cycles were summarized and visualized. On the basis of the results (measurements at 28 points in each pile), visualizations of the

spatial distribution of the gas concentrations (CO, $CO_2$, $O_2$ and NO) and temperature distribution in the piles was performed using the Surfer 10 software by Golden Software (estimation based on created value nets) and an example of the results is shown in Figure 1. Four cross-sections and two longitudinal cross-sections (left and right side of the pile) were made for each of the six piles.

**Supplementary Materials:** **File 1:** Table S1.xlsx contains Table S1. Summary of O2, CO, CO2, NO concentration and temperature within the composting process.; **File 2:** Figures.zip contains files (Figures S1–S230.pdf) with a graphical visualization of temperature and gases distribution during composting. Figure captions are included in the Figure-captions-Data.docx file.

**Author Contributions:** Conceptualization, A.B., S.S., G.P.; methodology, A.B., S.S, G.P.; formal analysis, A.B., J.K.; validation, A.B., G.P. J.K.; investigation, S.S. K.S. resources, K.S., S.S., data curation, A.B., K.S.; writing—original draft preparation, K.S., S.S., writing—review and editing, A.B., G.P., J.K.; visualization, K.S., supervision, A.B., J.K.

**Funding:** This work was supported by the Best-Eko Sp. z o.o. (Poland) as the research program 'Selection of substrates based on BEST-TERRA compost and composting technology at the composting plant at Boguszowice sewage treatment plant', No. B090/0010/17. Authors would like to thank the Fulbright Foundation for funding the project titled "Research on pollutants emission from carbonized refuse derived fuel into the environment", completed at Iowa State University. In addition, the preparation of this paper was partially supported by the Iowa Agriculture and Home Economics Experiment Station, Ames, Iowa. Project no. IOW05556 (Future Challenges in Animal Production Systems: Seeking Solutions through Focused Facilitation) sponsored by Hatch Act and State of Iowa funds.

**Conflicts of Interest:** The authors declare no conflict of interest. The funders had no role in the design of the study; in the collection, analyses, or interpretation of data; in the writing of the manuscript; or in the decision to publish the results.

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
