# Peer review of "The Spatial and Temporal Distribution of Process Gases within the Biowaste Compost"

_data_

Round 1
Reviewer 1 Report
The purpose of paper is clear. The collected data are complete and exhaustive.
Table 1: why indoor tests with turning regime of two times a week was not investigated?
Lines 124, please indicate days and not weeks.
Author Response
We uploaded the detail response in the attached file.

Reviewer 2 Report
The Data Descriptor entitled “The spatial and temporal distribution of process gases within the biowaste compost” summarizes the spatial and temporal distribution of process gases generated inside full-scale composting piles. The dataset is well-designed, however there are some points which must be improved. Authors must work on the following points:
-Abstract: Abstract should be rewritten by detailing the aim and concept of the dataset. The abstract should state briefly the purpose of the dataset, the principal results and major conclusions.
-Summary: Very general and need to be elaborative to explore the actual philosophy to design the dataset. The summary is insufficient to provide the state of the art in the dataset. Hypothesis should be given. How this dataset is different from the available data?
- Data Description: The data description section is well written and some 230 figures are presented as supplementary materials.
- Methods: Page 3, Line no 103-Add model and origin of instruments used for temperature and gas study here.
- Some related lines of future research and summarizing lines should be added.
Author Response

(The authors gave the same response as above.)

Reviewer 3 Report
The manuscript falls within the scope of the journal and represents a very important and actual topic.I think that manuscript can be published after following corrections:
- Please ensure more detail in Data description section.
- Please ensure more explanation for figures where that is possible.
- Delete titles for figures from manuscript which are supplementary materials. I don't saw that such titles are presented in other manuscript which are published in this journal.
- At the end of manuscript ensure discussion, user note or limitations.
- Please include more references.
Author Response

(The authors gave the same response as above.)

Round 2
Reviewer 2 Report
The authors made all the modifications to improve the dataset, increasing its quality. Therefore, the manuscript is accepted.